# Neural Language of Thought Models

**Yi-Fu Wu**[1]**, Minseung Lee**[2]**, Sungjin Ahn**[2]*
[1]Rutgers University [2]KAIST

## Abstract

The Language of Thought Hypothesis suggests that human cognition operates on a structured, language-like system of mental representations. While neural language models can naturally benefit from the compositional structure inherently and explicitly expressed in language data, learning such representations from non-linguistic general observations, like images, remains a challenge. In this work, we introduce the Neural Language of Thought Model (NLoTM), a novel approach for unsupervised learning of LoTH-inspired representation and generation. NLoTM comprises two key components: (1) the Semantic Vector-Quantized Variational Autoencoder, which learns hierarchical, composable discrete representations aligned with objects and their properties, and (2) the Autoregressive LoT Prior, an autoregressive transformer that learns to generate semantic concept tokens compositionally, capturing the underlying data distribution. We evaluate NLoTM on several 2D and 3D image datasets, demonstrating superior performance in downstream tasks, out-of-distribution generalization, and image generation quality compared to patch-based VQ-VAE and continuous object-centric representations. Our work presents a significant step towards creating neural networks exhibiting more human-like understanding by developing LoT-like representations and offers insights into the intersection of cognitive science and machine learning.

## 1 Introduction

The Language of Thought Hypothesis (LoTH) (Fodor et al., 1975) suggests that human cognition is based on a structured, language-like system of mental representations, often referred to as "Mentalese". Mentalese comprises word-like units that form sentence-like structures, which convey meaning. The meaning of these mental "sentences" is systematically determined by the meanings of their constituent "words" and their specific arrangement. From a computational viewpoint, while neural language models (Bengio et al., 2000; Brown et al., 2020; Bommasani et al., 2021) can benefit from the compositional and symbolic structure inherently expressed in the language data they are trained on, it remains unclear how we can learn such LoT-like structure from non-linguistic general observations, such as images, videos, and audio signals. The significance of this ability is further highlighted by the fact that infants learn these structures from observing objects and events before they acquire language skills (Spelke, 2022).

*How can we create neural networks that learn to develop such language of thought representations in an unsupervised way?* To address this, we outline the following three properties as the desired characteristics of a neural language of thought model.

First, when perceiving a visual scene, humans do not simply represent it as a monolithic vector of features. Instead, we view the scene structurally and semantically, recognizing it as a *composition of meaningful components* such as objects and their attributes, including shape, color, and position (Palmer, 1977; Singer, 2007; Spelke & Kinzler, 2007). Our observation here is that in line with the LoTH, these visual attributes can be likened to words, objects to sentences, and the scene to a paragraph. Recent works, particularly those focused on object-centric representations (Greff et al., 2020), have demonstrated that this structural decomposition facilitates the benefits associated with the LoTH such as relational reasoning (Wu et al., 2021; Yoon et al., 2023; Webb et al., 2023a;b) and out-of-distribution generalization (Dittadi et al., 2022; Yoon et al., 2023) due to increased compositional generalization.

---

*Correspondence to sungjin.ahn@kaist.ac.kr

**Table 1:** Desiderata for Neural Language of Thought Models and Related Models

|  | VAE | VQ-VAE | Slot Attention | SysBinder | NLoTM (Ours) |
|---|---|---|---|---|---|
| **Compositionality** (Semantic Scene Decomposition) | Factor | ✘ | Object | Object & Factor | Object & Factor |
| **Symbolic** (Discrete Concept Abstraction) | ✘ | ✓(Patch Concept) | ✘ | ✘ | ✓(Semantic Concept) |
| **Productivity** (Probabilistic Compositional Generation) | ✓ | ✓(Patch Stitching) | ✘ | ✘ | ✓(Semantic Composition) |

Moreover, these structured and semantic representations can be categorized and conceptualized, resulting in *symbol-like discrete concept abstraction*. Such an ability is critical for organizing and comprehending the complexity of the environment, e.g., via language, as well as for implementing modularity (Andreas et al., 2016) or symbolic reasoning (Lake et al., 2016). Such discrete representations are also useful to leverage powerful generative models like autoregressive transformers (Vaswani et al., 2017). One of the most popular models for discrete representation learning is VQ-VAE (van den Oord et al., 2017). It has been shown to be beneficial for image generation (Razavi et al., 2019; Esser et al., 2021) and probability density modeling (Van den Oord et al., 2016). However, VQ-VAE and its variants, such as dVAE (Ramesh et al., 2021; Singh et al., 2022a) and VQ-GAN (Esser et al., 2021), represent a scene as a grid of small patches, lacking the capability to capture the scene's holistic structure and semantics.

Besides, the ability to *compositionally and probabilistically generate samples* that adhere to the distribution of prior beliefs, constructed from observation data, is crucial for endowing AI with the capabilities to imagine and simulate. These abilities are essential for tasks such as planning (Mattar & Lengyel, 2022; Hafner et al., 2019) and reasoning. However, this ability, related to probabilistic Language of Thought (PLoT) (Goodman et al., 2015) and productivity, is supported by only a certain class of representation learning models. While models like Slot Attention (Locatello et al., 2020) and SysBinder (Singh et al., 2023) offer structured, object-centric compositional representations, in their original form it is unclear how they support density-based sampling. In contrast, VAE-based models support this ability to sample from a prior distribution. However, they either do not provide object-centric structures or are limited to patch-based discrete abstractions (VQ-VAE).

In this work, we present the Neural Language of Thought Model (NLoTM), the first model that satisfies the aforementioned criteria summarized in Table 1. NLoTM comprises two novel components: the Semantic Vector-Quantized (SVQ) Variational Autoencoder and the Autoregressive LoT Prior (ALP). SVQ achieves discrete semantic decomposition of a scene by learning hierarchical, composable factors that closely align with the objects and the properties of objects in visual scenes, similar to the role of words and sentences in LoTH. ALP, analogous to language models trained on text tokens, is an autoregressive transformer trained to learn a probabilistic generation of semantic concept tokens. However, unlike VQ-VAE which stitches a grid of patches, ALP composes semantic concepts such as objects and their attributes to represent a scene.

In our experiments, we demonstrate the following practical benefits of our method. First, we find that for multi-object scenes, NLoTM is able to model the prior distribution better than patch-based methods, as measured by the quality of the samples generated. Second, we find that NLoTM representations outperform patch-based VQ representations on downstream tasks that require knowledge of the different properties of the objects in the scene. We also find evidence that NLoTM representations can generalize better to out-of-distribution tasks compared to patch-based VQ representations and SysBinder continuous representations. Lastly, we show that despite introducing a discrete bottleneck, NLoTM can work on the challenging CLEVRTex (Karazija et al., 2021) dataset, one of the most complex datasets used in recent unsupervised object-centric representation learning models.

Our contributions are as follows: First, we introduce NLoTM, the first neural network model implementing the LoTH for unstructured observations. Second, we propose the SVQ model to obtain object-centric semantic neural discrete representations. Third, we propose the ALP to model the probabilistic prior capable of capturing the underlying data distribution and compositional generation of new samples. Lastly, we evaluate our model on several 2D and 3D datasets including the challenging CLEVRTex dataset, showing superior downstream task performance and image generation quality.

## 2 BACKGROUND

### 2.1 VECTOR-QUANTIZED VARIATIONAL AUTOENCODER (VQ-VAE)

The VQ-VAE (van den Oord et al., 2017) is a model that learns to compress high-dimensional data into a discretized latent space. The latent space is maintained by a codebook of prototype vectors $\mathbf{e} \in \mathbb{R}^{K \times d}$ where $K$ is the size of the codebook and $d$ is the dimensionality of each prototype vector. An input $\mathbf{x}$ is first passed through encoder $E(\mathbf{x})$ to obtain latents $\mathbf{z}_e \in \mathbb{R}^d$. A nearest-neighbor lookup between $\mathbf{z}_e$ and each of the prototype vectors in the codebook yields a quantized representation $\mathbf{z}_q = \text{Quantize}(\mathbf{z}_e) = \mathbf{e}_k$ where $k = \arg\min_j ||\mathbf{z}_e - \mathbf{e}_j||_2$. The decoder $D$ then uses $\mathbf{z}_q$ to reconstruct the input: $\hat{\mathbf{x}} = D(\mathbf{z}_q)$. The model is trained with the following loss:

$$\mathcal{L} = \underbrace{||\mathbf{x} - \hat{\mathbf{x}}||_2^2}_{\text{Reconstruction}} + \underbrace{||sg[\mathbf{z}_e] - \mathbf{z}_q||_2^2}_{\text{Codebook}} + \beta \underbrace{||\mathbf{z}_e - sg[\mathbf{z}_q]||_2^2}_{\text{Commitment}} .$$

The first term is a reconstruction loss and is used to train the encoder and decoder. A straight-through estimator (Bengio et al., 2013) is used to estimate the gradients through the quantization step by copying the gradients from $\mathbf{z}_q$ to $\mathbf{z}_e$. The second term is the codebook loss which encourages the prototype vectors in the codebook to be close to the output of the encoder. The third term, scaled by a constant hyperparameter $\beta$, is the commitment loss and helps to stabilize the training by encouraging the output of the encoder to not deviate too much from the chosen prototype vectors. Instead of the codebook loss, we use exponential moving average (EMA) updates on the codebook, which we found to speed up training in our experiments (Razavi et al., 2019; Dhariwal et al., 2020; Yan et al., 2021).

When VQ-VAEs are applied to images $\mathbf{x} \in \mathbb{R}^{H \times W \times C}$, the encoder $E(\mathbf{x})$ is typically implemented as a convolution encoder, outputting a feature map of latents $\mathbf{z}_e \in \mathbb{R}^{H_z \times W_z \times d}$. This means that each latent corresponds to a local area represented by a convolutional feature cell and thus can only capture information in a local receptive field (Figure 1a). However, images typically contain multiple objects, and the discrete factors underlying visual scenes typically correspond to different properties of the objects in the scene, such as shape, color, type, and so on. The local patches from convolutional feature maps are inadequate to capture this rich structure.

### 2.2 OBJECT-CENTRIC REPRESENTATIONS

The goal of unsupervised object-centric representation learning is to decompose a scene into a set of representations each capturing a different object in the scene. It is shown that this structural decomposition, matching to the true factor structure of the world, facilitates some high-level cognition abilities such as relational reasoning (Wu et al., 2021; Yoon et al., 2023; Webb et al., 2023a;b) and out-of-distribution generalization (Dittadi et al., 2022; Yoon et al., 2023). We build on top of Slot Attention (Locatello et al., 2020), a spatial attention-based object-centric representation method.

Given an image $\mathbf{x} \in \mathbb{R}^{H \times W \times C}$, slot attention learns a set of slots, $\mathbf{s} = \{\mathbf{s}_1, \ldots, \mathbf{s}_N\}$, where $\mathbf{s}_n \in \mathbb{R}^{d_s}$ and $N$ is the total number of slots. An encoder is applied to $\mathbf{x}$ and, after adding a positional encoding, the result is flattened to an $L$-length input feature vector $\mathbf{F} \in \mathbb{R}^{L \times d_F}$. Then, an iterative attention mechanism is used to spatially group the input features $\mathbf{F}$ to the slot representations $\mathbf{s}$. First, the slots are randomly initialized from a Gaussian distribution with learned parameters. Then, in each iteration, the slots are used as queries in an *inverted* version of dot-product attention (Tsai et al., 2020) with the input features $\mathbf{F}$ as the keys and values. Instead of normalizing over the keys as is done in traditional dot-product attention, normalization is done over the queries (ie. slots). Additionally, a weighted mean is used to aggregate the values instead of the normal weighted sum, which is shown to stabilize training. The result is then used to update the slots with a per-slot GRU (Chung et al., 2014) followed by a per-slot residual MLP, both with shared parameters across the slots.

The slot representations are then used in a decoder to reconstruct the image and the entire model is trained with an image reconstruction loss. The original formulation of slot attention used a spatial broadcast decoder (Watters et al., 2019b) to create masked images per slot which are then combined to form a final reconstructed image. Recently, (Singh et al., 2022a) proposed using a transformer decoder to reconstruct the image while attending to the slots with cross attention. This method was shown to scale to more complex scenes than the spatial broadcast decoder (Singh et al., 2022b) and is what we choose to use in our model.

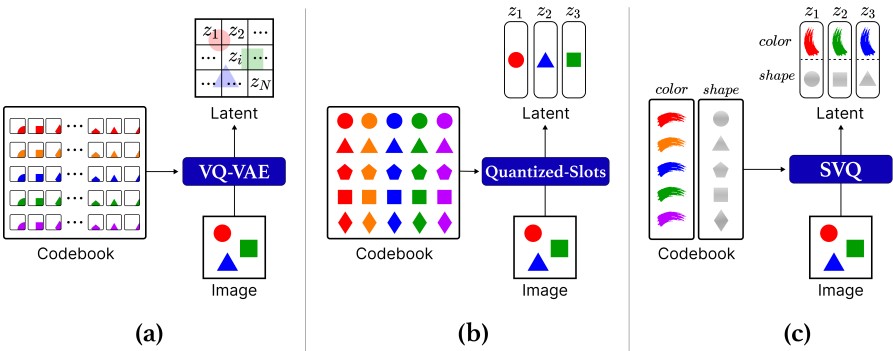

**Figure 1:** Comparison between VQ-VAE, Quantized Slots, and SVQ. (a) VQ-VAE quantizes the scene at a local patch level and may not capture the semantic structure of the scene. (b) Quantized Slots (QS) would quantize the scene at the slot level but require a separate code for every possible configuration of an object. (c) SVQ quantizes at the block level, representing each factor (such as color or shape) as a code. In this example, to represent all possible object configurations, SVQ requires only 10 codebook entries at the block level while QS requires 25.

## 3 NEURAL LANGUAGE OF THOUGHT MODEL

### 3.1 SEMANTIC VECTOR QUANTIZATION

Given a slot attention encoder that can obtain a set of representations of the objects in a scene, one may think of a hypothetical method, applying vector quantization to the slot representation itself to obtain a set of semantic discrete representations (Figure 1b). While these representations would indeed correspond to the different objects in a scene, this scheme would require one codebook entry per possible object configuration and be insufficient for anything beyond trivially simple scenes.

For example, consider a simple scene containing a single object in a fixed position that only varies by color and shape. Assume there are $c$ possible colors and $s$ possible shapes for the object. With slot-level quantization, in order to represent all the potential objects, the codebook would require at least $c \times s$ entries. This is because each slot representation is a single entangled representation, so each combination of factors needs to be represented by a separate code. If, instead, we were able to disentangle the object-level representations into factor-level representations—representations that align with the underlying latent factors of variation of each object—we could describe the potentially large combinatorial space of each object with a much smaller number of discrete factors. In the above example, if we had a fully disentangled representation of the color and the shape, we could represent all possible scenes with $c + s$ codes (Figure 1c). See Appendix B.2 for further discussion.

This observation motivates us to design an architecture that further disentangles slot representations to factor representations that reflect the underlying discrete factors of the objects in the scene, and to perform vector quantization on these factor representations. Under this scheme, each object representation would be composed of multiple discrete factors, and each factor would have its own codebook that can be shared across objects. The resulting model, the **S**emantic **V**ector-**Q**uantized Variational Autoencoder (SVQ), is depicted in Figure 2a and described below.

To obtain factored representations, we follow an approach motivated by Neural Systematic Binder (SysBinder) (Singh et al., 2023), where a binding mechanism is introduced to produce disentangled factors within a slot. Specifically, the following modifications are applied to slot attention: First, we maintain $M$ codebooks $\mathbf{C} \in \mathbb{R}^{M \times K \times d_c}$, each with $K$ discrete prototype vectors of dimension $d_c = \frac{d_s}{M}$. Then, we split each of the $N$ $d_s$-dimensional slot representations into $M$ equally-sized blocks, each of which will represent one factor. We denote the full set of block representations as $\mathbf{z}_e \in \mathbb{R}^{N \times M \times d_c}$. Crucially, we replace the slot-level GRUs and residual MLPs with block-level equivalents that have shared parameters across blocks corresponding to the same factor. At the end of each slot attention iteration, we apply vector quantization for each block using its corresponding codebook to obtain a set of quantized blocks $\mathbf{z}_q \in \mathbb{R}^{N \times M \times d_c}$. For $n \in [1, N], m \in [1, M]$,

$$\mathbf{z}_q^{n,m} = \mathbf{C}_{m,k} \text{ where } k = \arg\min_j ||\mathbf{z}_e^{n,m} - \mathbf{C}_{m,j}||_2 \,,$$

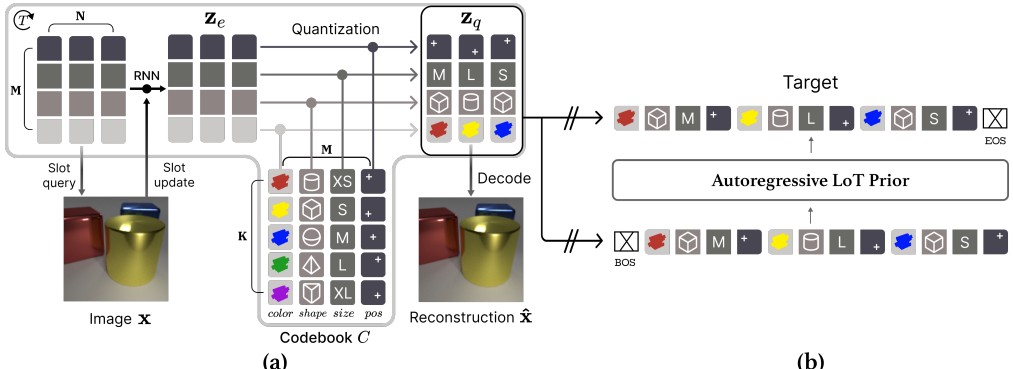

**Figure 2:** Overall architecture of NLoTM. (a) The Semantic Vector-Quantized (SVQ) Variational Autoencoder. We maintain $M$ learned codebooks and split each slot into $M$ blocks. After each Slot Attention iteration, we apply vector quantization to each block representation to obtain a set of discrete codes for each slot. Each block ends up specializing to different underlying factors of variation for the objects in the scene. (b) The Autoregressive LoT Prior (ALP). We train an autoregressive prior over the discrete latent codes from SVQ. Sampling from this prior allows us to generate an image one object at a time, based on their properties.

where $\mathbf{z}_q^{n,m}$ denotes the $m$-th block in the $n$-th slot and $\mathbf{C}_{m,k}$ is the $k$-th prototype vector in the $m$-th codebook. By sharing the codebook for each block across all of the slots, each block ends up specializing in different underlying factors of the objects in the scene, such as color, shape, and position. Thus, these quantized representations are semantic in the sense that they contain factor-level representations mapping to the underlying structure of the scene.

To reconstruct the image, we use the autoregressive transformer decoder described in Section 2.2 and condition on $\mathbf{z}_q$ via cross attention. Similar to Singh et al. (2023), we first let the blocks within a slot interact with a single-layer transformer and then add a block-level positional encoding before inputting the representations as cross attention in the transformer decoder. We train the model with the reconstruction loss, the VQ-VAE commitment loss, and we update the codebooks with EMA updates. To prevent codebook collapse, we also incorporate random restarts for the embeddings, similar to previous work (Dhariwal et al., 2020). To achieve this, we keep a count of the usage of each code in the codebooks and randomly reset it to be near one of the encoder outputs of the current batch if its usage falls below a threshold.

## 3.2 AUTOREGRESSIVE LANGUAGE OF THOUGHT PRIOR

Given these semantic discrete codes representing the different objects in the scene, we can now freeze the SVQ and train a prior $p(\mathbf{z}_q)$ over these codes to obtain a generative model of the underlying data that captures the structure and semantics of the data, analogous to language models that are trained over text tokens. We can sample from this prior, called Autoregressive LoT Prior (ALP), to obtain codes for new scenes and use these codes in the SVQ decoder to generate new images. Compared to patch-based VQ methods that generate patch tokens corresponding to a local region of an image, ALP generates a representation of a scene one object at a time, based on their properties (Figure 2b).

We implement the prior using a simple autoregressive transformer decoder. First, we flatten $\mathbf{z}_q$ along the slot and block dimensions to a vector with dimensions $NM \times d_c$. We then apply a positional encoding across all slots and blocks and input the resulting vector to a transformer decoder with an objective of predicting the discrete code of the next block. Although slot attention does not guarantee any specific ordering of the slots, the blocks within the slots are arranged in a predefined order. Therefore, the positional encoding is important in providing information about the ordering of the blocks as well as which block belongs to which slot.

Note that generating the latents of one image requires sampling $NM$ blocks, but does not depend on the size of the image. This is different than VQ-VAE, which scales with the size of the feature map and may become expensive for high-resolution images.

# 4 RELATED WORK

**Neural Discrete Representation Learning**. Our work builds on top of neural discrete representation learning, which has played a pivotal role in the advancement of generative models for images in recent years (van den Oord et al., 2017; Razavi et al., 2019; Ramesh et al., 2021; Esser et al., 2021; Yu et al., 2022). These methods typically follow a two-stage approach. First, an image is encoded into a CNN feature map, which is then tokenized using vector quantization (Gray, 1984) into a set of discrete latent variables. In the second stage, a powerful autoregressive prior is then trained to model the distribution of these discrete tokens, allowing for sampling new images from this distribution. Our model also follows this two-stage approach, except our latents correspond to the properties of objects instead of cells in a CNN feature map.

**Unsupervised Object-Centric Learning**. Recent unsupervised object-centric learning methods have been shown to decompose an image or video into a set of latents, each representing an object in the scene (Burgess et al., 2019; Greff et al., 2019; Anciukevicius et al., 2020; Locatello et al., 2020; Greff et al., 2017; Engelcke et al., 2020; 2022; von Kügelgen et al., 2020; Du et al., 2021; Kabra et al., 2021; Zhang et al., 2022; Eslami et al., 2016; Lin et al., 2020b; Jiang & Ahn, 2020; Chen et al., 2021; Deng et al., 2021; Lin et al., 2020b;a; Singh et al., 2023; Kipf et al., 2022; Singh et al., 2022b; Gopalakrishnan et al., 2022; Seitzer et al., 2022; Hénaff et al., 2022; Wang et al., 2023a; Wu et al., 2021; Wen et al., 2022; Zoran et al., 2021). While most of these methods result in a distributed representation per object, there have been several attempts at learning more structured or disentangled representations, such as those methods that decompose the latents into what and where components (Eslami et al., 2016; Crawford & Pineau, 2019b;a; Jiang et al., 2019; Jiang & Ahn, 2020; Lin et al., 2020b;a; Chen et al., 2021) or those that learn disentangled latents via a VAE (Greff et al., 2019; Zoran et al., 2021). Closely related to our work, recent methods have been designed to learn factor-level disentanglement (Singh et al., 2023; Kirilenko et al., 2023). However, these methods still operate with continuous latents instead of discrete tokens and do not support sampling new images. While there are several object-centric learning methods that do support sampling new images (Engelcke et al., 2020; 2022; Jiang & Ahn, 2020; Wang et al., 2023b), these also do not use semantic discrete latents as we do in our work.

# 5 EXPERIMENTS

**Datasets.** We evaluate our model on two variants of a 2D Sprites dataset (Watters et al., 2019a; Yoon et al., 2023) and three variants of the CLEVR dataset (Johnson et al., 2017), CLEVR-Easy, CLEVR-Hard, CLEVR-Tex. In the 2D Sprites datasets, objects of varying shapes and colors are placed in a scene. In total, there are 7 possible colors and 12 possible shapes. In each image, one object has a single property that is unique from the other objects. All other properties are shared by at least two objects. This structure allows us to evaluate if the prior correctly models the dependencies between the properties of the scene. We test versions of this dataset with and without textured backgrounds (Cimpoi et al., 2014). CLEVR-Easy, CLEVR-Hard, and CLEVR-Tex were previously used in (Singh et al., 2023) and are modified from the original CLEVR (Johnson et al., 2017) and CLEVR-Tex (Karazija et al., 2021) datasets to have larger objects so properties such as shape and texture are more clearly visible. In CLEVR-Easy, objects may differ by only shape, color, and position. In this dataset, there are 3 possible shapes and 8 possible colors. In CLEVR-Hard, objects may differ by shape, color, position, size, and material. There are 3 possible shapes, 137 possible colors, and 2 possible materials (shiny or matte). In CLEVR-Tex, there are 4 possible shapes and 58 possible textures for the objects and background.

**Baselines.** We compare our model with several patch-based quantization methods: VQ-VAE (van den Oord et al., 2017) with a PixelCNN (Van den Oord et al., 2016) prior, and dVAE (Ramesh et al., 2021; Singh et al., 2022a) with a transformer decoder prior. For the dVAE baseline, we use the dVAE weights that are trained along with the SVQ. This provides a more direct ablation comparing the ALP of NLoTM with the patch-based transformer decoder prior since the dVAE decoder is shared across these models and will not contribute to differences in image quality. We also compare with GENESIS-v2 (Engelcke et al., 2022), a continuous latent object-centric model with an autoregressive prior that can also generate samples.

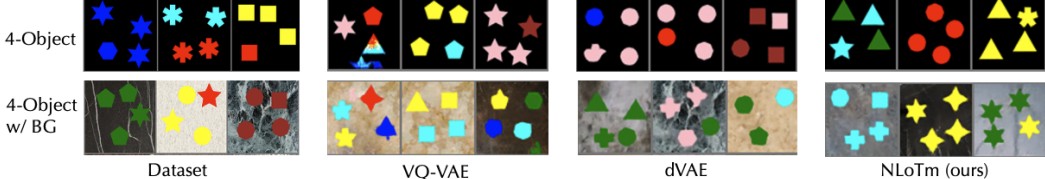

**Figure 3:** Generated samples for the 4-object 2D Sprites and 4-object 2D Sprites with background datasets.

## 5.1 GENERATING SAMPLES WITH THE AUTOREGRESSIVE LOT PRIOR

### 5.1.1 2D SPRITES

We show the sample generations for the 2D Sprites datasets in Figure 3 and the FID results in Table 2. We additionally calculate generation accuracy by manually inspecting 128 images per model to check if the generated images follow the constraints of the dataset. That is, each image must have exactly one object that has a unique property. All other properties in the scene will have at least one duplicate among the other objects.

We see that for the simplest dataset with 3 objects and no background, NLoTM achieves the lowest FID of the models and comparable generation accuracy to dVAE, generating about 75% of the scenes correctly. This setting may be simple enough that dVAE with a transformer prior can capture the structure of the scene even with a patch-based discrete latent. As the scene complexity increases with more objects and textured background, NLoTM starts to outperform the baselines in terms of generation accuracy. Inspecting the qualitative results, we see that in the dataset with the background, VQ-VAE and dVAE start generating occasional blurry objects, whereas NLoTM maintains clean-looking objects that match the ground truth dataset. This may be because NLoTM can segment the background into its own slot and factor the texture into a discrete latent, cleanly separating the representation of the objects from the background. The patch-based methods, however, may have a harder time separating the foreground from the background resulting in messier generations. Interestingly, despite the blurry shapes, VQ-VAE achieves the lowest FID score on the 2D Sprites dataset with background. We hypothesize this may be because the model spends more capacity modeling the background correctly instead of the foreground, which may produce a better FID score, but not necessarily better generation accuracy. This is confirmed by the low generation accuracy of the VQ-VAE model this dataset, only generating 19.5% of the scenes correctly.

**Table 2:** FID and Generation Accuracy on the 2D Sprites datasets. For Generation Accuracy, 128 samples were inspected manually to determine if they matched the constraints of the scene (ie. exactly one unique property among all the shapes). Underlined numbers indicate a minor difference from the best value.

| | FID ↓ | | | Generation Accuracy (in %) ↑ | | |
|---|---|---|---|---|---|---|
| Dataset | VQ-VAE | dVAE | NLoTM (ours) | VQ-VAE | dVAE | NLoTM (ours) |
| 2D Sprites (3 obj) | 14.81 | 7.26 | **6.61** | 28.91 | **75.78** | 75.00 |
| 2D Sprites (4 obj) | 26.35 | 19.15 | **17.93** | 21.88 | 62.50 | **66.41** |
| 2D Sprites w/ bg (4 obj) | **58.14** | 66.08 | 58.50 | 19.53 | 30.47 | **42.19** |

### 5.1.2 CLEVR

In Figure 4, we show sample generations after training the models on the CLEVR-Easy, CLEVR-Hard, and CLEVR-Tex datasets. We report the Frechet Inception Distance (FID) in Table 3. We find that compared to the other models, GENESIS-v2 generates very blurry images and completely fails on CLEVR-Tex, resulting in a high FID. While VQ-VAE produces sharper images, several of the generated shapes are malformed or have mixed colors. The dVAE-generated images look closer to the ground truth dataset, but still have some errors such as overlapping objects (first image) and generating scenes with more objects than seen in the training set (third image). NLoTM has the lowest FID for all of these datasets and the generated images look very close to the ground truth dataset, indicating the usefulness of the ASP for generating these multi-object scenes.

In Appendix B.1, we show additional analysis of the learned codebook on the CLEVR-Easy dataset.

**Figure 4:** Generated samples for the CLEVR-Easy, CLEVR-Hard, and CLEVR-Tex Datasets.

**Table 3:** FID for the various models on the CLEVR datasets.

| Dataset | FID ↓ | | | |
|---|---|---|---|---|
| | GENESIS-v2 | VQ-VAE | dVAE | NLoTM (ours) |
| CLEVR-Easy | 115.56 | 57.06 | 40.30 | **32.50** |
| CLEVR-Hard | 93.01 | 73.33 | 65.89 | **43.12** |
| CLEVR-Tex | 225.08 | 178.59 | 112.80 | **84.52** |

## 5.2 DOWNSTREAM TASKS

### 5.2.1 ODD-ONE-OUT

We first evaluate on a downstream supervised learning task on the 2D Sprites dataset. We modify the dataset by first dividing each image into four quadrants and ensuring exactly one object will be in each quadrant. As in our previous experiments, one object has a single property that is unique from the other objects. The goal of the task is to identify the quadrant of the odd-one-out object. We first pretrain the baseline models on a dataset containing all 12 possible shapes and 7 possible colors. Then, we freeze the underlying model and train a downstream model on top of the learned representations with the supervised objective. The downstream model is trained on a dataset that only contains 9 possible shapes and 4 possible colors. We then evaluate on both the in-distribution (ID) dataset and an out-of-distribution (OOD) dataset that consists of the remaining 3 shapes and 3 colors. In addition to dVAE and VQ-VAE, we use SysBinder as a baseline for this task, to compare its continuous representation with NLoTM's discrete representation. For the latent representation of NLoTM, we include variants that use the codebook indices in the SVQ (NLoTM Indices) and the codebook prototype vectors in the SVQ (NLoTM Codebook).

Table 4 shows the results of our experiments. Since all models can solve the task when evaluated on the ID dataset, we report the number of steps to reach 98% accuracy on the validation dataset. We find that SysBinder and NLoTM Codebook learn the quickest in the ID setting. For the OOD setting, we find that dVAE and VQ-VAE fail, not performing better than randomly guessing, showing that the patch-based discrete latent is insufficient for OOD generalization in this task. SysBinder can partially solve the task in the OOD setting, while the NLoTM Codebook seems to be able to solve the task, achieving 99% accuracy. This indicates that the compact latent space offered by the discrete code provides better OOD generalization abilities for this particular task. One possible explanation for this is that since this is an odd-one-out task, the downstream network needs to do comparisons between the properties of the objects and this may be easier to do with SVQ's codebook vectors that are fixed. SysBinder's continuous latents, on the other hand, offer greater variations for the same concept. This

**Table 4:** Results for the downstream odd-one-out task. Since all the models can solve the in-distribution (ID) task, we report the number of steps to 98% ID Accuracy and out-of-distribution (OOD) accuracy.

|  | Steps to 98% (↓) | OOD Acc. % (↑) |
|---|---|---|
| dVAE Discrete | 37,000 | 26.7 |
| dVAE Continuous | 32,000 | 29.5 |
| VQ-VAE Indices | 77,000 | 24.0 |
| VQ-VAE Codebook | 54,500 | 55.6 |
| SysBinder | **27,000** | 67.6 |
| NLoTM Indices | 77,000 | 46.8 |
| NLoTM Codebook | **27,000** | **99.1** |

**Table 5:** Results for the downstream CLEVR-Hard Property Comparison task.

|  | ID Acc. % (↑) | OOD Acc. % (↑) |
|---|---|---|
| dVAE Discrete | 27.52 | 19.87 |
| dVAE Continuous | 24.51 | 20.51 |
| VQ-VAE Indices | 24.53 | 17.74 |
| VQ-VAE Codebook | 23.73 | 18.80 |
| SysBinder | **79.60** | 70.09 |
| NLoTM Indices | 68.21 | 64.53 |
| NLoTM Codebook | 75.86 | **71.15** |

increases the potential for the downstream network to learn spurious correlations in the data, which can negatively impact OOD performance. NLoTM Indices is also only able to partially solve the task. This makes sense because in the out-of-distribution case, the model does not have any way of knowing two codebook indices are for the same property value (e.g. if two codebook vectors both correspond to the color blue). Since NLoTM Codebook uses the prototype vectors, it does not have this problem because the similarity can be determined by the vector representation.

### 5.2.2 CLEVR-HARD PROPERTY COMPARISON

For CLEVR-Hard, we construct a downstream task that assigns a number to each image as follows: First, we assign a number for each possible shape, color, and material present in the dataset. Then, for a given image, we identify the maximum number for each of these three properties. Lastly, we sum the max numbers for each of the properties to arrive at one integer label per image. We formulate the problem as a classification problem to correctly identify the number for each image. For example, suppose we have a scene containing a matte red cylinder and a shiny blue sphere. Assume we assign the following numbers to the different property values: matte $= 0$, shiny $= 1$, red $= 5$, blue $= 3$, cylinder $= 4$, sphere $= 6$. Thus the two objects are represented by the numbers $(0, 5, 4)$ and $(1, 3, 6)$. The max numbers for each of the properties is $(1, 5, 6)$ and the final integer label is $1 + 5 + 6 = 12$. Solving this task requires understanding the property values of each object in the scene.

We train the underlying models on the entire dataset consisting of all the possible property values. Then we randomly select 50 objects for an OOD dataset. Since our task relies on knowing the numerical value of each property, the ID dataset we train on may still contain property values of objects in the out-of-distribution dataset, but it will not contain objects where the combination of property values is present in the OOD dataset. Thus, when evaluating on the OOD dataset, we are testing the model on novel *combinations* of property values, even if those property values were individually observed during training. We show the ID and OOD results in Table 5. We see that SVQ outperforms the patch-based methods and performs comparably to SysBinder in both ID and OOD settings. This shows that despite adding a discretization bottleneck, the latents in SVQ are still useful for downstream tasks that rely on the properties of the objects in the scene.

## 6 CONCLUSION

In this paper, we introduce the Neural Language of Thought Model (NLoTM). This is the first model that satisfies our proposed desiderata of a neural network model of the LoTH: (1) Compositional semantic representation, (2) Symbol-like discrete concept abstraction, (3) Probabilistic neural grammar to generate samples compositionally. The NLoTM consists of two main components: Semantic Vector Quantized Variational Autoencoder, which learns hierarchical and composable discrete representations aligned with objects and their properties; and Autoregressive Language of Thought Prior, which learns to generate semantic concept tokens compositionally in a probabilistic manner. We evaluated NLoTM on several 2D and 3D image datasets. It demonstrated superior performance in downstream tasks, out-of-distribution generalization, and image generation quality compared to patch-based VQ-VAE and continuous object-centric representations. By further developing and refining models like NLoTM, we believe that we can continue to make progress in creating AI systems that exhibit more human-like understanding and generalization capabilities.

## ETHICS STATEMENT

The scope of our study was restricted to visually simple, procedurally generated scenes and in its current form does not pose any immediate ethical concerns. Future work, however, that extends the capabilities of our model to work on more complex scenes may have the potential to generate fake, realistic-looking images. The semantic discrete latent may allow users to control scenes in ways that were not previously explored. While this may serve to enhance productivity, such as for artists and graphic designers, it could also be used maliciously in the hands of a bad actor. Future researchers pursuing this direction should do so under strong ethical standards and be cognizant of the potential misuse of this technology.

## REPRODUCIBILITY STATEMENT

In addition to details about our model described in Section 3.1, we provide additional implementation details in Appendix C, including detailed hyperparameters used in our experiments. We will also release the source code upon acceptance of the paper.

## ACKNOWLEDGMENTS

This work is supported by Brain Pool Plus Program (No. 2021H1D3A2A03103645) and Young Researcher Program (No. 2022R1C1C1009443) through the National Research Foundation of Korea (NRF) funded by the Ministry of Science and ICT. We thank Gautam Singh for insightful discussions and help with the CLEVR datasets. We also thank Sjoerd van Steenkiste for valuable feedback on an earlier draft of this paper.

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

## A    LIMITATIONS

While our method can learn semantic discrete representations and is capable of using these representations to generate images of higher visual fidelity than previous object-centric methods such as GENESIS (Engelcke et al., 2020; 2022), it is still only shown to work well on synthetic datasets with similar visual complexity as previous work (Singh et al., 2023). Although scaling unsupervised object-centric models to more realistic datasets is not a focus of this work, further improving our model so that it can work well on more realistic scenes is an important avenue of future research. Another limitation of our model is that our latent representations are *all* discrete. Although our visual world does consist of many discrete concepts, factors such as position and pose are continuous. It would be interesting to explore ways to combine continuous and discrete factors to better model realistic scenes.

## B    ADDITIONAL EXPERIMENTAL RESULTS

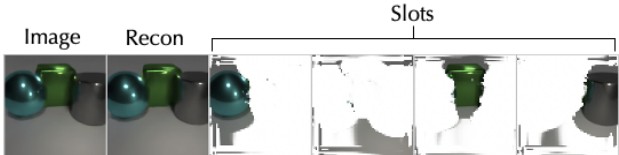

**Figure 5:** Sample scene we use in our codebook analysis.

### B.1    CODEBOOK ANALYSIS

**Latent Traversal.** In this section, we qualitatively analyze the codebook for a sample scene. Figure 5 shows the sample we will use in our analysis. First, we run the image through the pretrained SVQ encoder to obtain a set of semantic discrete latents. Each latent represents one block from one slot and is provided by a prototype vector in the corresponding codebook for that block. To investigate the effect of traversing through the codebook, we replace each block with a different code in the codebook while keeping all other latents fixed. We then reconstruct the scene with the SVQ decoder and dVAE, essentially generating a new image that only differs from the original image by one discrete latent.

Figure 6 shows the results for several sample blocks for the first slot (which corresponds to the teal ball) and the fourth slot (which corresponds to the gray cylinder). For each block, we choose the same set of 16 prototype vectors to display. First, we see that the slots are disentangled at the object level—changing one block in one slot does not affect the other objects. We also see that the different blocks specialize in different factors. Block 1 corresponds to the left and right placement of the object. Block 3 also corresponds to the placement of the object, but seems to also control the forward and backward placement of the object, as well as the size of the object. We notice that in this particular case, the factors of position and size are not completely disentangled. This may be because in this scene, the size depends on the placement of the object (e.g. closer objects are bigger). Block 7 controls the color of the object. We see that the same prototype vector seems to produce the same color, although there are some inconsistencies such as the disappearing cylinder in the bottom left. The color also seems to be cleanly disentangled from the other factors—changing the color does not affect other factors like shape, size, or position.

**Block Analysis.** Next, to further explore the representation captured in the codebook, we visualize the objects that are attended to for different prototype vectors. To achieve this, we run the pretrained SVQ on 1000 images obtaining the semantic discrete latents and slot attention segmentation maps for the objects in the images. Then, for each prototype vector in the codebook, we find and visualize the corresponding slots that are utilizing that code in one of its blocks. Note that unlike Singh et al. (2023), we do not need to do any k-means clustering to obtain this visualization since our representations are discrete representations in the codebook. Figures 7 and 8 show sample objects corresponding to three different prototype vectors for block 3 and block 7. We see that block 3 corresponds to object size and block 7 corresponds to object color. These results are consistent with the previous latent traversal

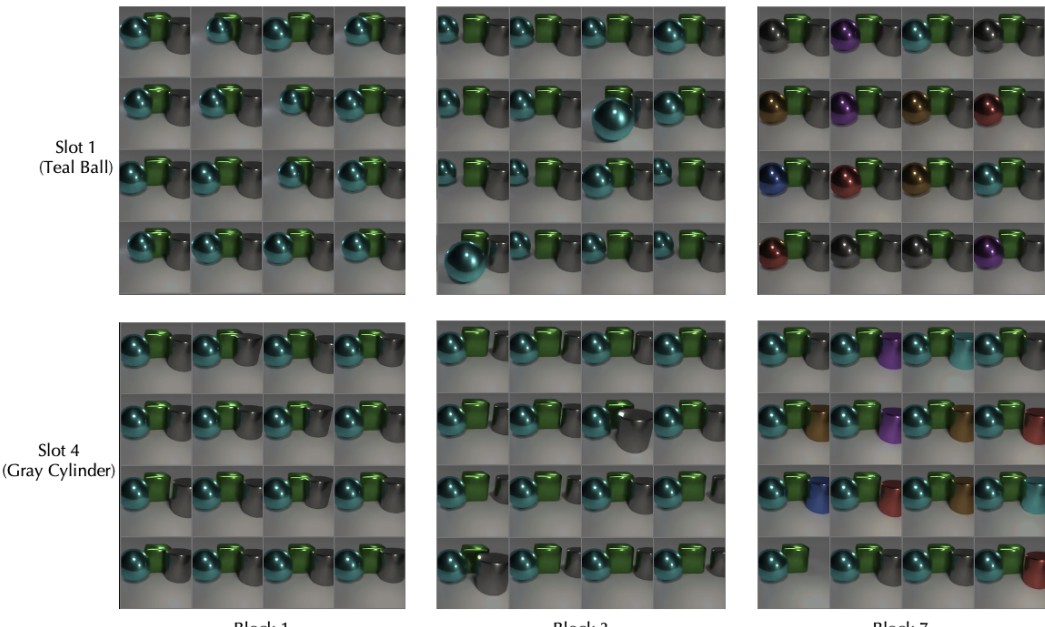

**Figure 6:** Latent traversal changing one latent in one block at a time while keeping all other latents fixed. The image is then reconstructed with the single changed latent.

experiments. Furthermore, the three prototype vectors we chose for block 7 correspond with the first three latents in Figure 6 (right), showing that these three prototype vectors represent gray, purple, and teal, respectively.

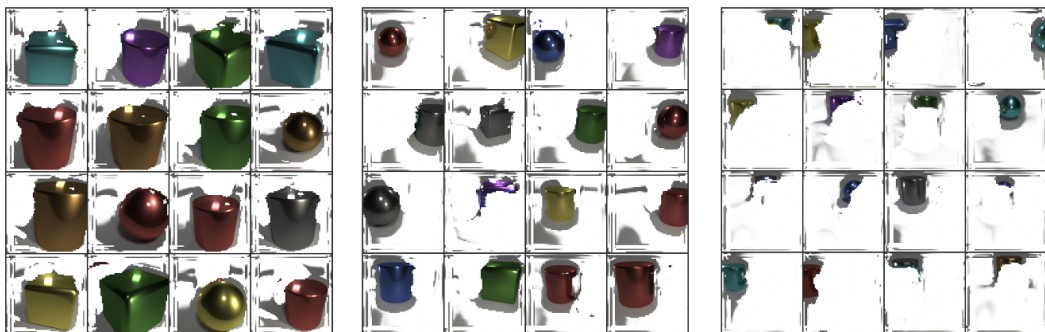

**Figure 7:** Objects attended to when the latent for block 3 is set to three different prototype vectors.

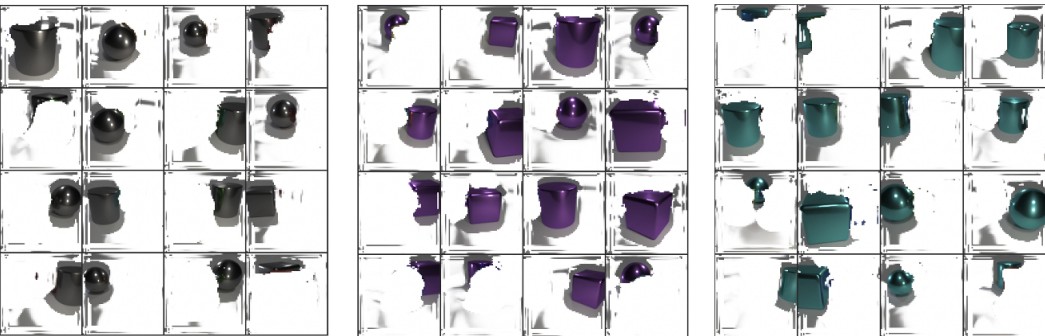

**Figure 8:** Objects attended to when the latent for block 7 is set to three different prototype vectors.

## B.2 Comparison with Slot-Level Quantization

As discussed in Section 3.1, we hypothesize that slot-level discretization would struggle with complex scenes due to the combinatorial nature of the underlying factors of the objects. We test this hypothesis by running experiments on 2D Sprites and CLEVR-Easy where we set the number of blocks $M$ to 1 and tune the size of the codebook, essentially doing slot-level quantization. In Figures 9, we show the masked attention of each slot on the input image as well as the image reconstruction. We find that with slot-level quantization, the model completely fails on the CLEVR-Easy dataset, unable to cleanly attend to the objects and reconstruct the image. On the 2D sprites dataset, we see that with slot discretization, one slot ends up attending to all the foreground objects and the model still cannot reconstruct the input image correctly. These results point to the importance of our choice to do block-level discretization.

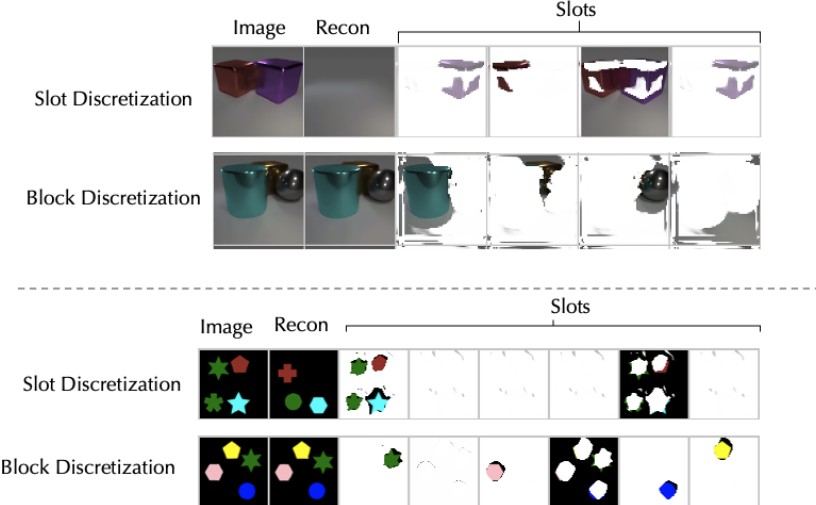

**Figure 9:** Comparison of slot discretization and block discretization on CLEVR-Easy (top) and 2D sprites (bottom).

## B.3 Prior Model Capacity for dVAE

In order to evaluate whether or not a larger capacity prior may improve the results for the patch-based dVAE baseline, we ran ablations using larger transformers for the dVAE prior on the CLEVR-Hard dataset. The results are presented in Table 6. We see that while increasing the size of the transformer for the prior does slightly improve the FID for dVAE, it still underperforms when compared to NLoTM, indicating that simply scaling the dVAE prior may not be sufficient to match NLoTM performance.

**Table 6:** Effect of increasing the dVAE prior model capacity on FID on the CLEVR-Hard dataset.

| Prior Model | FID |
|---|---|
| dVAE (8-layer) | 65.89 |
| dVAE (12-layer) | 61.74 |
| dVAE (16-layer) | 60.75 |
| NLoTM (8-layer) | 43.12 |

## B.4 Number of Blocks Ablation

Table 7 shows the results of changing the number of blocks in SVQ on the 2D Sprites (3 obj) dataset. We see that when the number of blocks is too small, the model performs poorly and fails to generate

scenes corresponding to the data distribution. For a sufficiently large number of blocks, the model is able to segment the scene, but Generation Accuracy decreases when the number of blocks is too large. We suspect this to be because with more blocks, the model may require a higher capacity prior, which we kept fixed in this ablation.

**Table 7:** Effect of changing the number of blocks in SVQ on the 2D Sprites (3 obj) dataset.

| Number of Blocks | FID | Generation Accuracy (in %) |
|---|---|---|
| 1 | 465.60 | 0.00 |
| 2 | 80.71 | 1.56 |
| 4 | 7.76 | 75.78 |
| 8 | 6.61 | 75.00 |
| 16 | 7.17 | 55.47 |
| 32 | 8.74 | 54.69 |

### B.5 CODEBOOK SIZE ABLATION

Table 8 shows the results of changing the codebook size in SVQ on the 2D Sprites (3 obj) dataset. We had also tried smaller codebook sizes of 4 and 16, but found that for codebook sizes smaller than 32, the SVQ model did not converge well, resulting in black reconstructions. Similar to the number of blocks ablations, we see that for larger codebook sizes, the FID scores are similar, but generation accuracy decreases for codebook sizes larger than 64. This again may be because the larger codebook sizes require a higher capacity prior, which was fixed in these ablations.

**Table 8:** Effect of changing the codebook size in SVQ on the 2D Sprites (3 obj) dataset.

| Codebook Size | FID | Generation Accuracy (in %) |
|---|---|---|
| 32 | 7.31 | 76.56 |
| 64 | 6.61 | 75.00 |
| 96 | 6.64 | 51.56 |
| 128 | 7.58 | 48.44 |
| 256 | 8.73 | 32.81 |

### B.6 FG-ARI SEGMENTATION RESULTS

Table 9 shows the Foreground Adjusted-Rand-Index (FG-ARI) metric for the CLEVR datasets for different slot-based models. We see that when compared to SysBinder, NLoTM performs similarly in terms of FG-ARI on CLEVR-Easy and CLEVR-Hard and slightly underperforms on CLEVR-Tex. Compared to vanilla Slot Attention, NLoTM achieves higher FG-ARI on all 3 datasets.

**Table 9:** FG-ARI results on CLEVR datasets.

| | Slot Attention | SLATE | SysBinder | NLoTM |
|---|---|---|---|---|
| CLEVR | 85.85 | 91.65 | 92.58 | 91.37 |
| CLEVR-Hard | 81.29 | 76.79 | 90.43 | 90.48 |
| CLEVR-Tex | 24.67 | 73.85 | 78.12 | 70.93 |

### B.7 EXPERIMENTS ON GOOGLE SCANNED OBJECTS

In order to evaluate NLoTM on a more realistic dataset, we use a dataset where the objects are taken from the Google Scanned Objects (Downs et al., 2022). Specifically, we use the objects from the "Shoe" and "Bottles and Cans and Cups" categories. We evaluate on two versions of NLoTM: NLoTM-small uses the same hyperparameters as we used for the CLEVR-Hard dataset (see Table

11) and NLoTM-large increases the codebook size to 256 and increases the size of ALP model from 8 layers, 4 heads, model size 192 to 16 layers, 8 heads, model size 512.

We present the FID results in Table 10 and qualitative samples in Figure 10. We see that while NLoTM-small is able to generate objects from the dataset, the objects are smoothed out, resulting in a high FID score. Increasing the model size to NLoTM-large significantly improves the quality of the generated scenes and the FID score, providing some evidence that NLoTM can be scaled to work on more realistic datasets.

**Table 10:** FID for the Google Scanned Objects dataset.

| Model | FID |
|---|---|
| NLoTM-small | 114.16 |
| NLoTM-large | 72.68 |

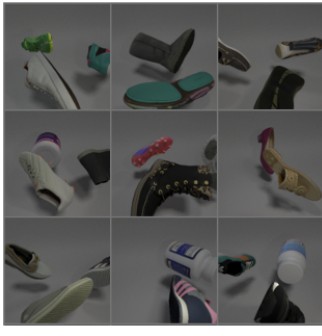 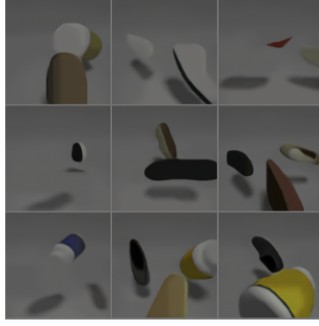 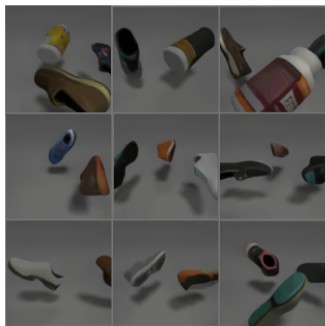

Dataset  NLoTM-small  NLoTM-large

**Figure 10:** Samples for NLoTM on the Google Scanned Objects dataset. Scaling to a larger model size noticeably improves the quality of the samples.

## C  IMPLEMENTATION DETAILS

### C.1  TRAINING AND IMPLEMENTATION DETAILS.

We use input images of 64x64 resolution for the 2D Sprites datasets and 128x128 for the CLEVR datasets. Each model is trained on NVIDIA Quadro RTX 8000 GPUs with 48GB memory and we use half-precision floating-point format. We train SVQ for 400k iterations which takes around 80 hours for the CLEVR datasets and 50 hours for the 2D datasets. We then train the ALP prior for 1 million iterations which takes around 40 hours. For the 2D Sprites dataset, similar to (Yoon et al., 2023), we first train the underlying models on a dataset of random shapes without any relationship between the objects. We then train the prior models on the odd-one-out datasets.

### C.2  HYPERPARAMETERS

Table 11 shows the hyperparameters we used for the different datasets in our experiments with SVQ. For the dVAE and Transformer Decoder, we follow the hyperparameters, architecture, and training procedure provided in Singh et al. (2023) for CLEVR-Easy and CLEVR-Hard. For the 2D Sprites datasets, we use the same hyperparameters as we do for CLEVR-Easy for those components. All models are trained with the Adam optimizer (Kingma & Ba, 2015) with $\beta_1 = 0.9$ and $\beta_2 = 0.999$.

### C.3  PRIOR MODELS

For the DVAE and ALP prior models, we use a transformer architecture with 8 layers, 4 heads, model dimension 192, feedforward dimension 768, and a dropout probability of 0.1. We use a learning rate

| Module | Hyperparameter | Dataset | | | |
|---|---|---|---|---|---|
| | | CLEVR-Easy | CLEVR-Hard | 2D Sprites | 2D Sprites w/ BG |
| General | Batch Size | 40 | 40 | 40 | 40 |
| | Training Steps | 400K | 400K | 400K | 400K |
| | Image Size | $128 \times 128$ | $128 \times 128$ | $64 \times 64$ | $64 \times 64$ |
| SVQ | Codebook Dimension | 256 | 128 | 256 | 32 |
| | # Blocks | 8 | 16 | 8 | 8 |
| | Codebook Size | 64 | 64 | 64 | 128 |
| | # Iterations | 3 | 3 | 3 | 3 |
| | # Slots | 4 | 4 | 6 | 8 |
| | $\beta$ | 50 | 50 | 50 | 50 |
| | Learning Rate | 0.0001 | 0.0001 | 0.0001 | 0.0001 |

**Table 11:** Hyperparameters of our model used in our experiments.

of 0.0003 and 30,000 warmup steps. For VQ-VAE, we use a 20-layers PixelCNN prior, as proposed in the original paper (van den Oord et al., 2017).

## C.4 DOWNSTREAM MODELS

For the 2D Sprites downstream experiments, we use a transformer architecture with 3 layers, 8 heads, model dimension 192, feedforward dimension 768, and a dropout probability of 0.1 for all models. We use the Adam optimizer with a learning rate of 0.0003.

For the CLEVR-Hard downstream experiments, we use a transformer architecture with 8 layers, 4 heads, model dimension 192, feedforward dimension 768, and a dropout probability of 0.1 for all models. We use the Adam optimizer with a learning rate of 0.0001.

