# OpenReview forum: "Neural Language of Thought Models"
_ICLR.cc/2024/Conference — ICLR 2024 poster_

### Official Review · Reviewer_41Cm · 2023-10-31

**Soundness:** 2 fair
**Presentation:** 2 fair
**Contribution:** 2 fair
**Rating:** 6
**Confidence:** 3

**Summary:**

This paper presents a modification of VQ-VAE called Semantic Vector-Quantized Variational Autoencoder (SVQ), which provides semantic discrete object representation by learning block-level factors, leveraging advances in object-centric learning. Different from patch-based VAE models, SVQ first applies slot attention to obtain object-level codebooks, then splits the slots into blocks following a binding mechanism to represent semantic factors. In comparison with previous methods, the model demonstrates better image sampling quality considering object fidelity. The proposed representation also shows promising performance on several downstream tasks in the out-of-distribution setting.

**Strengths:**

（1）The paper is overall well-written and the idea of learning semantic representation is well-motivated.
（2）Extensive and thorough experiments have been carried out to verify the effectiveness of the approach on multiple datasets and settings.

**Weaknesses:**

Weaknesses:
（1）	Novelty: The major limitation of this paper is the lack of theoretical novelty, since it seems the idea of disentangling object-level slots into blocked-level factors as well as the autoregressive transformer decoder for image reconstruction have been introduced in prior work SysBinder. It’s not elaborated on how the proposed approach is developed further in representing discrete semantics, especially when there’s not a large margin between the performance of SVG and other baselines.
（2）	Semantic illustration: Throughout the paper (including the name of the proposed model), the authors emphasize that the model is capable of learning semantic representation of objects such as shapes, colors, etc. However, how and what semantics are actually learned with unsupervised learning in the codebook is never illustrated or visualized, which makes the statement less convincing.
（3）	Experiment: First, the specification of multiple hyper-parameters (e.g. number of blocks, dimension of vectors) of SVG and other implementation details is never found. Second, the artificial designs of the constraints of the scene (metric Generation Accuracy) as well as the odd-one-out downstream task are not explained. Third, as the authors mentioned in the section of Limitations, the evaluations are restricted within synthetic datasets with simple scenarios. It’s unclear whether the proposed method will work in realistic, complex datasets, especially when the representation is object-related whose effectiveness is likely to be related to the size of the codebook.

**Questions:**

My questions are listed in weakness part.

---

> ### Author Response · Authors · 2023-11-20
> **Response to Reviewer 41Cm**
>
> Thank you for the review! We kindly refer the reviewer to our supplementary material which we believe should clarify several of your questions.
>
> > Novelty: The major limitation of this paper is the lack of theoretical novelty, since it seems the idea of disentangling object-level slots into blocked-level factors as well as the autoregressive transformer decoder for image reconstruction have been introduced in prior work SysBinder. It’s not elaborated on how the proposed approach is developed further in representing discrete semantics, especially when there’s not a large margin between the performance of SVG and other baselines.
> >
>
> While our work builds on the foundation of SysBinder, it introduces significant and crucial extensions. Unlike SysBinder, which lacks the abilities to model prior distributions and perform sampling, we observe that by discretizing the block representations, we can train a novel semantic prior transformer. This enables us to achieve the **first semantic and composable modeling of representations and distributions from raw images**, which can be seen as an advancement towards learning a semantic language of visual scenes, similar to words in text.
>
> We want to clarify that while SysBinder and SVQ both use a transformer decoder for reconstruction, SVQ additionally is able to train a separate autoregressive transformer decoder as the Semantic Prior on top of the discrete latents (Section 3.1 and Figure 2b), allowing for sampling of new scenes. **This ability is not supported by SysBinder and is a major contribution of our work and a key motivation for obtaining discrete latents.** Our generated samples are shown to be better than several baselines in terms of quantitative metrics (FID and generation accuracy) as well as qualitatively as shown in Figures 3 and 4 and discussed in Section 5.1.
>
> > Semantic illustration: Throughout the paper (including the name of the proposed model), the authors emphasize that the model is capable of learning semantic representation of objects such as shapes, colors, etc. However, how and what semantics are actually learned with unsupervised learning in the codebook is never illustrated or visualized, which makes the statement less convincing.
> >
>
> **We actually provide this result** in the section A.1 of the Appendix in the supplementary material where we analyze the codebook for a sample scene (”Latent Traversal” section) and show the representations captured in the codebook (”Block Analysis”).
>
> > First, the specification of multiple hyper-parameters (e.g. number of blocks, dimension of vectors) of SVG and other implementation details is never found.
> >
>
> We also already provide the hyperparameters and additional training and implementation details of our experiments in section B in the Appendix in the supplementary material.
>
> > Second, the artificial designs of the constraints of the scene (metric Generation Accuracy) as well as the odd-one-out downstream task are not explained.
> >
>
> In the beginning of Section 5 in the Dataset section, we describe the 2D Sprites dataset:
>
> ”In the 2D Sprites datasets, objects of varying shapes and colors are placed in a scene. In total, there are 7 possible colors and 12 possible shapes. In each image, one object has a single property that is unique from the other objects. All other properties are shared by at least two objects. This structure allows us to evaluate if the prior correctly models the dependencies between the properties of the scene.”
>
> We then describe at the beginning of section 5.1.1 the Generation Accuracy metric:
>
> ”We additionally calculate generation accuracy by manually inspecting 128 images per model to check if the generated images follow the constraints of the dataset. That is, each image must have exactly one object that has a unique property. All other properties in the scene will have at least one duplicate among the other objects.”
>
> For the downstream task, we describe in section 5.2.1:
>
> ”We modify the dataset by first dividing each image into four quadrants and ensuring exactly one object will be in each quadrant. As in our previous experiments, one object has a single property that is unique from the other objects. The goal of the task is to identify the quadrant of the odd-one-out object.”
>
> Please let us know if there is still any confusion about this metric and task based on our description in the submitted manuscript. We would be glad to revise the text to make this more clear.
>
> > Third, as the authors mentioned in the section of Limitations, the evaluations are restricted within synthetic datasets with simple scenarios. It’s unclear whether the proposed method will work in realistic, complex datasets, especially when the representation is object-related whose effectiveness is likely to be related to the size of the codebook.
> >
>
> Please see the above common response for a discussion about the synthetic nature of the datasets.

---

> > ### Author Response · Authors · 2023-11-23
> > **Additional Experiments on Google Scanned Objects Dataset**
> >
> > In order to evaluate SVQ on a more realistic dataset, we use a dataset where the objects are taken from the Google Scanned Objects [5]. We present the results in the updated supplementary material in Appendix A.7. We use two versions of SVQ. SVQ-small uses the same hyperparameters as we used for the CLEVR-Hard dataset (see Table 11 in Appendix) and SVQ-large increases the codebook size to 256 and increases the size of the transformer decoder from 8 layers, 4 heads, model size 192 to 16 layers, 8 heads, model size 512. We see that while SVQ-small is able to generate objects from the dataset, the objects are smoothed out, resulting in a high FID score. Increasing the model size to SVQ-large significantly improves the quality of the generated scenes and the FID score, providing some evidence that SVQ can be scaled to work on more realistic datasets.
> >
> > Additionally, since the rebuttal period is ending soon, we kindly ask the reviewer to check if our rebuttal addresses all of your concerns. Specifically, we want to emphasize that the concerns about the Semantic illustration and hyperparameters were in the original submitted version of the supplementary material and the odd-one-out downstream task is explained in the main text.

---

> > ### Comment · Reviewer_41Cm · 2023-11-23
> > **Thanks for the response.**
> >
> > Thanks for the detailed response from the authors. Most of my concerns have been addressed, so I improve my score to 6.
> > In the rebuttal, the authors also stated that "Unlike SysBinder, which lacks the abilities to model prior distributions and perform sampling, we observe that by discretizing the block representations, we can train a novel semantic prior transformer." It seems "discretization" is an advantage over SysBinder. However, in subsection 5.2.2 about the experimental results in Table 5,  the authors stated as "This shows that despite adding a discretization bottleneck, the latents in SVQ are still useful for downstream tasks that rely on the properties of the objects in the scene.". My concern is about discretization, which is an advantage or a bottleneck?

---

> > > ### Author Response · Authors · 2023-11-23
> > >
> > > Thank you for the support! With regards to discretization, it is a bottleneck in the sense that the representation is restricted to the vectors in the codebook. However, by discretizing we can train the semantic prior transformer over these discrete codes which enables sampling of new scenes. Therefore, it is *both* a bottleneck as well as an advantage in enabling this new ability. Additionally, despite this bottleneck, which one may think would affect the representation quality, our experiments show that the representations are still useful for downstream tasks.

---

### Official Review · Reviewer_qSSD · 2023-11-01

**Soundness:** 3 good
**Presentation:** 3 good
**Contribution:** 3 good
**Rating:** 6
**Confidence:** 4

**Summary:**

The paper proposes a method for decomposing scenes hierarchically from low level factors such as color and shape to objects. The method learns a prior over these concepts which allows for generating data from the data distribution which most object-centric papers cannot do. In the the experiments they show that the representation can be used to solve downstream tasks which require reasoning about the properties of different objects in the scene. Additionally they show that the inductive biases proposed by the work leads to better FID score compared to VQ-VAE and similar generative models.

**Strengths:**

- The paper proposes an interesting idea and the experimental setup for the downstream tasks are well done. Evaluating whether the model has learned underlying generative rules of learning is a neat and, to my knowledge, novel way of probing the model. I think the idea would be impactful and the community would be interested if it works for more realistic and larger scale datasets.

- The qualitative results in figure 6 where the color, position and size are disentangled are impressive even for a simple, synthetic dataset.

- The paper is well written and figures are illustrative of the main idea.

- Overall the experiment are well done, with reasonable baselines and clear analysis of the results.

**Weaknesses:**

- The experiments are only conducted on synthetic, simple datasets. The variants of CLEVR are procedurally generated based on disentangled factors such as shape, color, and texture so it's unclear if the factor based approach proposed by this work generalizes to more realistic scenes.

- The main metric used to evaluate the method and baseline is FID. This metric seems orthogonal to the goal of object-centric decomposition. Typically object-centric papers such as Slot-Attention measure segmentation with adjusted random index (ARI). I ask for clarification on why FID is the primary metric for evaluating object-centric models. I think if generation is the primary goal, then the work should compare to generative models and VQ-VAE on more realistic generative datasets.

- I would be happy to improve my score if my concerns are addressed, particularly if the method works on real, more complex datasets than CLEVR.

**Questions:**

- How sensitive are the results to the codebook size and number of blocks? It seems that for CLEVR we can a priori know these hyper-parameters, but for real-world or more complex datasets how would we set these?

- It seems natural to compare the ARI with slot-attention or a variant of slot-attention. Is there a reason why the method shouldn't be compared in this manner?

---

> ### Author Response · Authors · 2023-11-20
> **Response to Reviewer qSSD**
>
> Thank you for the review and the thoughtful questions!
>
> > The experiments are only conducted on synthetic, simple datasets. The variants of CLEVR are procedurally generated based on disentangled factors such as shape, color, and texture so it's unclear if the factor based approach proposed by this work generalizes to more realistic scenes.
> >
>
> Please see the above common response for a discussion about the synthetic nature of the datasets.
>
> > The main metric used to evaluate the method and baseline is FID. This metric seems orthogonal to the goal of object-centric decomposition. Typically object-centric papers such as Slot-Attention measure segmentation with adjusted random index (ARI). I ask for clarification on why FID is the primary metric for evaluating object-centric models. I think if generation is the primary goal, then the work should compare to generative models and VQ-VAE on more realistic generative datasets.
> >
> > …
> >
> > It seems natural to compare the ARI with slot-attention or a variant of slot-attention. Is there a reason why the method shouldn't be compared in this manner?
> >
>
> Thank you for this suggestion. It is true that as an object-centric learning method, it seems natural to include FG-ARI as an evaluation metric as is typically done in previous works. We originally did not include this because object segmentation was not a key focus of our work. Instead, we emphasized the ability to **generate new samples through a composition of high-level semantic concepts** as a key contribution and instead focused our main evaluations on FID, generation accuracy, and qualitative samples. Since our method adds an additional discrete bottleneck, we also do not expect SVQ to improve segmentation performance over previous methods. Nonetheless, we agree with the reviewer that it is important to include this metric to see the effect of the discrete bottleneck on segmentation performance. We will include these numbers in an updated version of the manuscript and summarize the results below. We see that when compared to SysBinder, SVQ performs similarly in terms of FG-ARI on CLEVR-Easy and CLEVR-Hard and slightly underperforms on CLEVR-Tex. Compared to vanilla Slot Attention, SVQ achieves higher FG-ARi on all 3 datasets. **Importantly, note that SVQ achieves this by incorporating the crucial abilities of prior distribution modeling and sampling, which are not present in Slot Attention, SLATE, or SysBinder.**
>
> |  | Slot Attention | SLATE | SysBinder | SVQ |
> | --- | --- | --- | --- | --- |
> | CLEVR-Easy | 85.85 | 91.65 | 92.58 | 91.37 |
> | CLEVR-Hard | 81.29 | 76.79 | 90.43 | 90.48 |
> | CLEVR-Tex | 24.67 | 73.85 | 78.12 | 70.93 |
>
> > How sensitive are the results to the codebook size and number of blocks? It seems that for CLEVR we can a priori know these hyper-parameters, but for real-world or more complex datasets how would we set these?
> >
>
> This is an important question. The codebook size and number of blocks are currently hyperparameters like the number of heads or convolution filters in other models. Our method does not require the true codebook size or number of blocks as it learns in an unsupervised way, but there exists a range of hyperparameters that make it work better. Making the model work on more complex real-world scenes is definitely an important future direction.
>
> To answer your question, we ran an ablation on the number of blocks for the 2D Sprites (3 obj) dataset. We are additionally running an ablation on the codebook size and will respond with the results once those experiments are finished running.
>
> The results for the ablation on number of blocks are shown in the table below. We see that when the number of blocks is too small, the model performs poorly and fails to generate scenes corresponding to the data distribution. For a sufficiently large number of blocks, the model is able to segment the scene, but Generation Accuracy decreases when the number of blocks is too large. We suspect this to be because with more blocks, the model may require a higher capacity prior, which we kept fixed in this ablation.
>
> | Number of Blocks | FID | Generation Accuracy (in %) |
> | --- | --- | --- |
> | 1 | 465.60 | 0.00 |
> | 2 | 80.71 | 1.56 |
> | 4 | 7.76 | 75.78 |
> | 8 | 6.61 | 75.00 |
> | 16 | 7.17 | 55.47 |
> | 32 | 8.74 | 54.69 |

---

> > ### Author Response · Authors · 2023-11-21
> > **Codebook Size Ablation**
> >
> > We’ve additionally ran ablations on the codebook size for the 2D Sprites (3 obj) dataset. We present the results below. We also tried smaller codebook sizes of 4 and 16, but found that for codebook sizes smaller than 32, the SVQ model did not converge well, resulting in black reconstructions. Similar to the number of blocks ablations, we see that for larger codebook sizes, the FID scores are similar, but generation accuracy decreases for codebook sizes larger than 64. This again may be because the larger codebook sizes require a higher capacity prior, which was fixed in these ablations.
> >
> > | Codebook Size | FID | Generation Accuracy (in %) |
> > | --- | --- | --- |
> > | 32 | 7.31 | 76.56 |
> > | 64 | 6.61 | 75.00 |
> > | 96 | 6.64 | 51.56 |
> > | 128 | 7.58 | 48.44 |
> > | 256 | 8.73 | 32.81 |

---

> > ### Comment · Reviewer_qSSD · 2023-11-23
> > **Reviewer Response**
> >
> > I thank the authors for their response. Most of my concerns have been addressed. While it would be nice to see this method used on real-world scenes or data, I realize that most current methods evaluate on the synthetic CLEVR variants. Overall I think the method and experiments are interesting, and the experimental setups such as odd-one-out are a novel way of probing a visual models understanding. I improve my score to 6.

---

> > > ### Author Response · Authors · 2023-11-23
> > > **Additional Experiments on Google Scanned Objects Dataset**
> > >
> > > Thank you for your support! We are glad to hear that we have addressed most of your concerns.
> > >
> > > We also ran an additional experiment on a more realistic dataset to address your concern about applying SVQ to more realistic datasets. We use a dataset where the objects are taken from the Google Scanned Objects [5]. We present the results in the updated supplementary material in Appendix A.7. We use two versions of SVQ. SVQ-small uses the same hyperparameters as we used for the CLEVR-Hard dataset (see Table 11 in Appendix) and SVQ-large increases the codebook size to 256 and increases the size of the transformer decoder from 8 layers, 4 heads, model size 192 to 16 layers, 8 heads, model size 512. We see that while SVQ-small is able to generate objects from the dataset, the objects are smoothed out, resulting in a high FID score. Increasing the model size to SVQ-large significantly improves the quality of the generated scenes and the FID score, providing some evidence that SVQ can be scaled to work on more realistic datasets.

---

### Official Review · Reviewer_gjT4 · 2023-11-01

**Soundness:** 3 good
**Presentation:** 3 good
**Contribution:** 3 good
**Rating:** 8
**Confidence:** 3

**Summary:**

This paper proposes an approach to perform semantic-based vector quantization and learn a corresponding codebook as well as a generative model under object-centric learning scenarios. It uses a slot-attention-based encoder to obtain N object-centric features for each image, where the hidden dimension is divided by M groups and each group shares the parameters. The M denotes the number of attributes. Further, it learns a codebook with MxK codes to quantize the features from the encoder. For every slot, each attribute would be quantized to the code with the closest L2 distance among the corresponding K codes, resulting in the final NxM quantized results for the reconstruction task. Experiments are conducted on various synthetic datasets to showcase the effectiveness of the proposed model. The results show that it is capable to capture high-level information beyond patch and generate better results.

**Strengths:**

The writing is clear and easy to follow.
The motivation looks good to me, and the proposed idea is interesting, the codebook analysis results are impressive. The performance looks good on these synthetic datasets.

**Weaknesses:**

I'm not up-to-date to the latest slot-attention models and their performance on synthetic datasets such as CLEVR. It seems that the authors only choose a few baselines (GENESIS-v2, VQ-VAE, dVAE) for the generation quality evaluation as well as other downstream baselines. The VQ-VAE/dVAE model may requires larger model capacity to handle this tasks as they works on the local patch level. The generation results in Fig.4 is surprisingly bad, given the fact that they actually perform well on real data generation (VQGAN/DALL-E), it should be easy to reconstruct/generate good results when the hyperparameters are appropriate. I'm not sure whether the current results is convincing enough.

I also have several questions in the following section, it would be great if the authors could address them. Overall I think this paper shows some interesting results and could inspire people. But the current experiments only show results on synthetic data which is somewhat weak.

**Questions:**

1. In Tab.4, the authors only compare with dVAE and VQ-VAE Indices, which makes the comparison unfair as SVQ Indices also perform bad. Why not the authors also showcase the VQ-VAE code (like Tab.5) as well as dVAE feature (using the code id to index the weight of the first layer in the decoder)?
2. The Tab.5 seems not a fair comparison as well. VQ-VAE / dVAE are not designed for the object-centric tasks so it is expected they will not perform well on this task, on the other hand, SysBinder shows comparable and even better performance on this task.
3. Following 3, the paper lacks of ablation studies, which makes the reader hard to understand which part plays the critical role and is useful to support the author's claim, for example, what would happen when M varies and becomes smaller than the actual number of attributes?
4. The authors have shown some interesting codebook analysis results, would be great if the authors could also showcase whether the learned codebook is composable and can perform controllable generation (other than based on reconstruction) and by solely manipulate the code.
5. Recently there is a paper present similar idea (group level quantization https://arxiv.org/abs/2309.15505), which may give the authors some inspiration as well, note that this is a concurrent work so I'm not asking the authors to compare with them.

---

> ### Author Response · Authors · 2023-11-20
> **Response to Reviewer gjT4 (Part 1/2)**
>
> Thank you for the review and insightful questions.
>
> > The VQ-VAE/dVAE model may requires larger model capacity to handle this tasks as they works on the local patch level.
> >
>
> For VQ-VAE, we had used the same 20-layer PixelCNN as proposed in the original paper. For dVAE, which is a more direct ablation of the semantic prior because it shares the same image decoder as SVQ, we had the same suspicion that perhaps a larger capacity decoder could improve the results. We therefore ran several experiments with larger capacity priors (12-layer and 16-layer transformers compared to 8-layer from our paper) on CLEVR-Hard to see the effect. We present the results below:
>
> | Model | FID |
> | --- | --- |
> | dVAE (8-layer) | 65.89 |
> | dVAE (12-layer) | 61.74 |
> | dVAE (16-layer) | 60.75 |
> | SVQ (8-layer) | 43.12 |
>
> We see that while increasing the size of the transformer for the prior does slightly improve the FID for dVAE, it still underperforms when compared to SVQ, indicating that the performance issue may not be solely due to model capacity.
>
> Nevertheless, we would like to emphasize that, although we use generation quality as one of the evaluation metrics, the **main contribution of the paper is not to propose a better image synthesizer but to answer the question “how can we learn semantic and composable representations (like language of visual thoughts) from raw images similar to the role of words in text.** Although patch-level quantized representation is known to be good for image generation, it does not provide semantic (or conceptual) and composable representation as we do.
>
> > In Tab.4, the authors only compare with dVAE and VQ-VAE Indices, which makes the comparison unfair as SVQ Indices also perform bad. Why not the authors also showcase the VQ-VAE code (like Tab.5) as well as dVAE feature (using the code id to index the weight of the first layer in the decoder)?
> >
>
> Thank you for this suggestion. As suggested, we’ve ran these additional experiments and will update the manuscript. We’ve reproduced the results below (new results highlighted in **********bold**********). We see that while the VQ-VAE Codebook performs better in the 2D Odd-One-Out experiments when compared with the other patch-based methods, **it still take more steps to reach 98% ID accuracy and has lower OOD Accuracy than SVQ.** The continuous version of dVAE does not seem to improve performance on the CLEVR-Hard downstream task. We will include these results in an updated version of the paper.
>
> 2D Odd-One-Out Downstream Experiment:
>
> | Model | Steps to 98% (ID) | OOD Accuracy % |
> | --- | --- | --- |
> | dVAE Discrete | 37,000 | 26.7 |
> | **dVAE Continuous** | 32,000 | 29.5 |
> | VQ-VAE Indices | 77,000 | 24.0 |
> | **VQ-VAE Codebook** | 54,500 | 55.6 |
> | SysBinder | 27,000 | 67.6 |
> | SVQ Indices | 77,000 | 46.8 |
> | SVQ Codebook | 27,000 | 99.1 |
>
> CLEVR-Hard Property Comparison Downstream Experiment:
>
> | Model | ID Accuracy % | OOD Accuracy % |
> | --- | --- | --- |
> | dVAE Discrete | 27.52 | 19.87 |
> | **dVAE Continuous** | 24.51 | 20.51 |
> | VQ-VAE Indices | 24.53 | 17.74 |
> | VQ-VAE Codebook | 23.73 | 18.80 |
> | SysBinder | 79.60 | 70.09 |
> | SVQ Indices | 68.21 | 64.53 |
> | SVQ Codebook | 75.86 | 71.15 |
>
>
> > The Tab.5 seems not a fair comparison as well. VQ-VAE / dVAE are not designed for the object-centric tasks so it is expected they will not perform well on this task, on the other hand, SysBinder shows comparable and even better performance on this task.
> >
>
> Here, we respectfully disagree. We believe that the experiment comparing SVQ with VQ-VAE and dVAE is fair and serves an important purpose. The VQ-VAE and dVAE baselines for the downstream task utilize an architecture similar to the Vision Transformer (ViT) architecture, where patch-codes are used as inputs and transformer layers are employed for predictions. The results highlight that the general ViT architecture struggles to handle reasoning tasks effectively, despite the presence of transformer layers. This outcome may not be immediately expected, as ViTs and their variants are widely recognized as powerful representation learners. We believe that this experiment is crucial in supporting the motivation behind our semantic representations that go beyond patch representations.
>
> It is worth noting that while SysBinder performs slightly better in the ID setting and slightly worse in the OOD setting compared to SVQ, it lacks the ability to provide a prior distribution and sampling. The comparison to SysBinder is, therefore, still crucial because it suggests that SVQ's representations remain effective even after incorporating prior modeling and sampling through the addition of the discrete bottleneck.

---

> > ### Author Response · Authors · 2023-11-20
> > **Response to Reviewer gjT4 (Part 2/2)**
> >
> > > Following 3, the paper lacks of ablation studies, which makes the reader hard to understand which part plays the critical role and is useful to support the author's claim, for example, what would happen when M varies and becomes smaller than the actual number of attributes?
> > >
> >
> > Thank you for this suggestion. We ran an ablation on the number of blocks for the 2D Sprites (3 obj) dataset. We are additionally running an ablation on the codebook size and will respond with the results once those experiments are finished running.
> >
> > The results for the ablation on number of blocks are shown in the table below. **We see that when the number of blocks is too small, the model performs poorly and fails to generate scenes corresponding to the data distribution.** For a sufficiently large number of blocks, the model is able to segment the scene, but Generation Accuracy decreases when the number of blocks is too large. We suspect this to be because with more blocks, the model may require a higher capacity prior, which we kept fixed in this ablation.
> >
> > | Number of Blocks | FID | Generation Accuracy (in %) |
> > | --- | --- | --- |
> > | 1 | 465.60 | 0.00 |
> > | 2 | 80.71 | 1.56 |
> > | 4 | 7.76 | 75.78 |
> > | 8 | 6.61 | 75.00 |
> > | 16 | 7.17 | 55.47 |
> > | 32 | 8.74 | 54.69 |
> >
> > > The authors have shown some interesting codebook analysis results, would be great if the authors could also showcase whether the learned codebook is composable and can perform controllable generation (other than based on reconstruction) and by solely manipulate the code.
> > >
> >
> > Thank you for this suggestion. One thing we’d like to point out is that **the Latent Traversal experiments (Figure 6) already provides controllable generation results**. Note that the results are created by modifying the code for one block while keeping the other blocks fixed. Thus, they are not based on reconstruction and can be seen as a form of controllable generation. For example, in Figure 6, if we look at the bottom left image of the middle column (Block 3), we see the effect of changing this block on Slot 1 (originally the teal ball) and Slot 4 (originally the gray cylinder). The effect is similar in both cases — the object is enlarged and moved closer to the camera.
> >
> > > Recently there is a paper present similar idea (group level quantization https://arxiv.org/abs/2309.15505), which may give the authors some inspiration as well, note that this is a concurrent work so I'm not asking the authors to compare with them.
> > >
> >
> > Thank you for pointing this out. We will update our related works to include this work and consider it for future iterations of the model.

---

> > > ### Author Response · Authors · 2023-11-21
> > > **Codebook Size Ablation**
> > >
> > > We’ve additionally ran ablations on the codebook size for the 2D Sprites (3 obj) dataset. We present the results below. We also tried smaller codebook sizes of 4 and 16, but found that for codebook sizes smaller than 32, the SVQ model did not converge well, resulting in black reconstructions. Similar to the number of blocks ablations, we see that for larger codebook sizes, the FID scores are similar, but generation accuracy decreases for codebook sizes larger than 64. This again may be because the larger codebook sizes require a higher capacity prior, which was fixed in these ablations.
> > >
> > > | Codebook Size | FID | Generation Accuracy (in %) |
> > > | --- | --- | --- |
> > > | 32 | 7.31 | 76.56 |
> > > | 64 | 6.61 | 75.00 |
> > > | 96 | 6.64 | 51.56 |
> > > | 128 | 7.58 | 48.44 |
> > > | 256 | 8.73 | 32.81 |

---

> > > > ### Author Response · Authors · 2023-11-23
> > > > **Additional Experiments on Google Scanned Objects Dataset**
> > > >
> > > > > Overall I think this paper shows some interesting results and could inspire people. But the current experiments only show results on synthetic data which is somewhat weak.
> > > > >
> > > >
> > > > In order to evaluate SVQ on a more realistic dataset, we use a dataset where the objects are taken from the Google Scanned Objects [5]. We present the results in the updated supplementary material in Appendix A.7. We use two versions of SVQ. SVQ-small uses the same hyperparameters as we used for the CLEVR-Hard dataset (see Table 11 in Appendix) and SVQ-large increases the codebook size to 256 and increases the size of the transformer decoder from 8 layers, 4 heads, model size 192 to 16 layers, 8 heads, model size 512. We see that while SVQ-small is able to generate objects from the dataset, the objects are smoothed out, resulting in a high FID score. Increasing the model size to SVQ-large significantly improves the quality of the generated scenes and the FID score, providing some evidence that SVQ can be scaled to work on more realistic datasets.
> > > >
> > > > Additionally, since the rebuttal period is ending soon, we kindly ask the reviewer to check if our rebuttal addresses all of your concerns.

---

### Official Review · Reviewer_cC8S · 2023-11-01

**Soundness:** 3 good
**Presentation:** 3 good
**Contribution:** 3 good
**Rating:** 6
**Confidence:** 2

**Summary:**

The paper proposed a novel image-quantized method, called semantic vector-quantized variational autoencoder. The SVQ constructs representations hierarchically from low-level discrete concept schemas to high-level object representation. The author conducts experiments on various 2D and 3D object-centric datasets and validates the effectiveness of the SVQ.

**Strengths:**

i) Compared with widely used VAE and VQ-VAE, SVQ models the discrete abstraction and object-level representations simultaneously.

ii) The proposed semantic prior based on discrete latent codes can be directly used in generation tasks, and the visualization results show it is superior compared with the baseline methods.

**Weaknesses:**

I am not an expert in the quantization area, my main works are related to self-supervised learning. So I want to ask some questions about the application of self-supervised learning.

i) The patch-level quantization method VQ-VAE can maintain the geometry structure. So it is widely used in mask-image-modeling methods, such as BEiT.  Can SVQ also maintain such a geometry structure?

ii) If the SVQ can be applied in the general 2D-image generation methods, such as stable diffusion? I think it is a promising application of the quantization method.

**Questions:**

Refer to the weakness

---

> ### Author Response · Authors · 2023-11-20
> **Response to Reviewer cC8S**
>
> Thank you for the comments and the positive review!
>
> > The patch-level quantization method VQ-VAE can maintain the geometry structure. So it is widely used in mask-image-modeling methods, such as BEiT. Can SVQ also maintain such a geometry structure?
>
> I’m not sure I completely understand what is meant by maintaining the geometry structure in this context, but the discrete latents in SVQ correspond to the different properties of the objects in the scene, rather than a fixed receptive field in the image. While it’s certainly possible to do a form of masked reconstruction training (and a potentially interesting future research direction), this would be very different than what is typically done in methods such as BEiT.
>
> > If the SVQ can be applied in the general 2D-image generation methods, such as stable diffusion? I think it is a promising application of the quantization method.
>
> Similar to the previous response, since the SVQ latents correspond to the objects in the scene, it seems that they are not directly applicable to methods such as stable diffusion. That being said, as we mention in the general response, combining diffusion-based approaches with our method may be a promising direction to scale our method to work on more realistic scenes.

---

> > ### Comment · Reviewer_cC8S · 2023-11-20
> >
> > Thanks for the explanation of my questions. I think it is an interesting work, and the technical seems sound. I will keep my original positive ratings.
> >
> > In addition, I think it is a promising area to apply the proposed method in the general diffusion method and mask-image-modeling pretraining area.
> >
> > P.S. I want to declare that I am not an expert in this area. The AC may consider other reviewers' opinions with higher weights.

---

### Author Response · Authors · 2023-11-20
**General Response**

Thank you to all the reviewers for taking the time to review our paper and provide valuable feedback and suggestions. We will respond to each reviewer individually below, but first wanted to give a general response regarding the synthetic nature of the datasets and the applicability of our method to more complex, real-world datasets.

While the datasets used in our experiments are synthetic, they are in line with current unsupervised object-centric methods, notably Neural Systematic Binder (NSB) [1], which is the closest to our work. Specifically, the CLEVRTex dataset has been particularly challenging for fully unsupervised object-centric learning methods.

We also want to emphasize that the focus of our work **is not** an investigation into scaling unsupervised object-centric models to realistic datasets. While we agree that figuring out how to get these models to work better on realistic datasets is a very important research effort, this work focuses on an *orthogonal* direction: whether or not it is possible to learn the **discrete compositional factors corresponding to the underlying factors of variation** of a scene in a **completely unsupervised** way and how these factors can **enable generation of object-centric scenes**.

We will also point out that there is some recent work tackling scaling unsupervised object-centric learning to more realistic scenes such as DINOSAUR [2] or diffusion-based approaches [3, 4]. Since these methods are orthogonal to ours (i.e., they do not have the structured distribution modeling ability as ours), we can potentially combine them to scale SVQ to more realistic scenes. For example, as is done in DINOSAUR, we can potentially leverage a frozen, pretrained backbone such as the DINO encoder for our input features and our reconstruction target. Similarly, incorporating a diffusion decoder into our model is also an option for getting SVQ to work on more complex datasets. While these are interesting directions, this is not the focus of our current work but interesting future directions.

[1] Singh, G., Kim, Y., and Ahn, S. Neural Systematic Binder.

[2] Seitzer, M., Horn, M., Zadaianchuk, A., Zietlow, D., Xiao, T., Simon-Gabriel, C.-J., He, T.,
Zhang, Z., Schölkopf, B., Brox, T., et al. Bridging the Gap to Real-World Object-Centric Learning.

[3] Jiang, J., Deng, F., Singh, G., and Ahn, S. Object-Centric Slot Diffusion.

[4] Wu, Z., Hu, J., Lu, W., Gilitschenski, I., and Garg, A. SlotDiffusion: Unsupervised Object-Centric Learning with Diffusion Models.

---

> ### Author Response · Authors · 2023-11-23
> **Updated General Response**
>
> ### Additional Experiments on Google Scanned Objects Dataset
>
> In order to evaluate SVQ on a more realistic dataset, we use a dataset where the objects are taken from the Google Scanned Objects [5]. We present the results in the updated supplementary material in Appendix A.7. We use two versions of SVQ. SVQ-small uses the same hyperparameters as we used for the CLEVR-Hard dataset (see Table 11 in Appendix) and SVQ-large increases the codebook size to 256 and increases the size of the transformer decoder from 8 layers, 4 heads, model size 192 to 16 layers, 8 heads, model size 512. We see that while SVQ-small is able to generate objects from the dataset, the objects are smoothed out, resulting in a high FID score. Increasing the model size to SVQ-large significantly improves the quality of the generated scenes and the FID score, providing some evidence that SVQ can be scaled to work on more realistic datasets.
>
> ### Summary of Revisions
>
> We have updated the manuscript and supplementary material with many of the suggestions from the reviewers. We summarize these revisions below:
>
> 1. Added additional experiments on the prior model capacity for dVAE (Appendix A.3), number of blocks ablation (Appendix A.4), codebook size ablation (Appendix A.5), FG-ARI segmentation results (Appendix A.6), and the Google Scanned Objects dataset (Appendix A.7).
> 2. Updated Table 4 and Table 5 to include dVAE continuous and VQ-VAE codebook.
>
> [5] Downs, Laura and Francis, Anthony and Koenig, Nate and Kinman, Brandon and Hickman, Ryan and Reymann, Krista and McHugh, Thomas B. and Vanhoucke, Vincent. Google Scanned Objects: A High-Quality Dataset of 3D Scanned Household Items

---

### Meta-Review · Area_Chair_222c · 2023-12-06

**Metareview:**

The paper proposes an approach for learning discrete object-centric representations from images. The method, when applied to fairly simplistic synthetic data, works, learns a (largely) disentangled representation, supports sampling of new scenes and OOD generalization on downstream tasks.

After the rebuttal and discussion, the reviewers are overall positive. Here are some key pros and cons

Pros:
1. A reasonable idea that works quite well. The method both learns good representations and supports sampling.
2. Fairly extensive experimental evaluation, in particular the disentanglement results
3. Well presented

Cons:
1. Key issue: only simple synthetic data (somewhat more realistic data added in rebuttal is nice, but the results there are very preliminary - in particular, it is unclear how disentangled the representation is there) - it's not clear (as also admitted by the authors in discussion) that the method would scale to real-world data
2. On downstream tasks, not necessarily better than baselines, e.g. sometimes worse than SysBinder
3. Suspiciously bad generation results of the baselines (e.g. Figure 4, Table 3)
4. The representation is discrete, which is clearly limited (e.g. for object poses)

Overall, even though the evaluation is somewhat limited, the method is new and somewhat works. so the paper can be of interest. Therefore, I recommend acceptance at this point.

**Justification For Why Not Higher Score:**

1. Key issue: only simple synthetic data (somewhat more realistic data added in rebuttal is nice, but the results there are very preliminary - in particular, it is unclear how disentangled the representation is there) - it's not clear (as also admitted by the authors in discussion) that the method would scale to real-world data
2. On downstream tasks, not necessarily better than baselines, e.g. sometimes worse than SysBinder
3. Suspiciously bad generation results of the baselines (e.g. Figure 4, Table 3)
4. The representation is discrete, which is clearly limited (e.g. for object poses)

**Justification For Why Not Lower Score:**

1. A reasonable idea that works quite well. The method both learns good representations and supports sampling.
2. Fairly extensive experimental evaluation, in particular the disentanglement results
3. Well presented

---

### Decision · Program_Chairs · 2024-01-16

Accept (poster)